# The successful reintroduction of African wild dogs (*Lycaon pictus*) to Gorongosa National Park, Mozambique

**Paola Bouley**[1◉]*, **Antonio Paulo**[1◉], **Mercia Angela**[1◉], **Cole Du Plessis**[2◉], **David G. Marneweck**[3◉]

**1** Department of Conservation, Gorongosa National Park, Mozambique, **2** Carnivore Conservation Programme, Endangered Wildlife Trust, Johannesburg, South Africa, **3** Department of Nature Conservation Management, Natural Resource Science and Management Cluster, Faculty of Science, Nelson Mandela University, George, South Africa

◉ These authors contributed equally to this work.
* seagoose1@protonmail.com

**Data Availability Statement:** The data underlying the results presented in the study are available from figshare (DOI: 10.6084/m9.figshare.13214699).

## Abstract

Large carnivores have experienced widespread extirpation and species are now threatened globally. The ecological impact of the loss of large carnivores has been prominent in Gorongosa National Park, Mozambique, after most were extirpated during the 1977–92 civil war. To remedy this, reintroductions are now being implemented in Gorongosa, initiating with endangered African wild dogs (*Lycaon pictus*), hereafter 'wild dogs'. We describe the first transboundary translocation and reintroduction of founding packs of wild dogs to Gorongosa over a 28-month study period and evaluate the success of the reintroduction based on five key indicator categories. We also assess how wild dog space use and diet influenced their success. We found that pre-release, artificial pack formation in holding enclosures aided group cohesion and alpha pair establishment. Post-release, we also observed natural pack formations as a result of multiple dispersal events. Founder and naturally formed packs produced pups in two of the three breeding seasons and packs successfully recruited pups. Survival rate for all wild dogs was 73% and all mortality events were from natural causes. Consequently, the population grew significantly over the study period. All indicators of success were fully achieved and this study documents the first successful reintroduction of wild dogs into a large, unfenced landscape in Mozambique and only the second on the continent. Potential mechanisms underlying these early successes were the avoidance of habitats intensively used by lions, dietary partitioning with lion, avoidance of human settlements, and Gorongosa's management strategy. We predict further population expansion in Gorongosa given that 68% of the park is still unused by wild dogs. This expansion could be stimulated by continued reintroductions over the short- to medium-term. Recovery of wild dogs in Gorongosa could aid in the re-establishment of a larger, connected population across the greater Gorongosa-Marromeu landscape.

**Funding:** The authors received no specific funding for this work.

**Competing interests:** The authors have declared that no competing interests exist.

## Introduction

Despite their ecological and socio-economic importance, large carnivores are currently highly threatened globally [1, 2]. Most species are listed as threatened or endangered with 77% still experiencing population declines due to anthropogenic drivers such as habitat loss [2, 3], depleted prey populations [4], and poaching and direct persecution by humans [1, 5, 6]. Some of the largest range contractions of carnivores have occurred on the African continent [3] where human population is projected to double to 2.2–2.5 billion by 2050 [7, 8]. Multiple conservation strategies are now urgently needed, including expansion of and strengthening management of functional ecological landscapes, sustainable human development around protected areas (especially poverty alleviation, a key driver of ecological degradation [9, 10]), establishing landscape connectivity and viable habitat corridors [11], and fostering human-carnivore coexistence [12, 13]. Where extirpations have occurred and where connectivity and natural recolonization is unlikely, reintroductions (i.e. conservation translocations) of species in to safe spaces remains a viable recovery tool [14–19].

The recently restored Gorongosa National Park (GNP), Mozambique's flagship park, represents an opportunity for carnivore recovery. Historically a renowned stronghold for wildlife, GNP was contemporaneous with the Serengeti and Ngorongoro Crater ecosystems and supported a high biomass of wildlife [20, 21]. Unfortunately, GNP was geographically at the epicenter of the 1977–92 civil-war, and this severe, war-driven perturbation resulted in a >90% decline in populations of large mammals and carnivores [22, 23] and ensuing significant imprints on ecological function [24]. In 2007, a long-term, public-private ecosystem restoration initiative—"the Gorongosa Restoration Project (GRP)"—was established with the goals of restoration and protection of wildlife populations, post-war recovery of the region's park-based economy, and improving surrounding communities' access to employment, education, health services, and sustainable agriculture [10]. By 2018 large herbivore populations had recovered by 95% [20] followed by the recovery of an extant, indigenous population of lions (*Panthera leo*) [25]. This multi-trophic level recovery as a result of strengthened management efforts, increasing interconnectivity of habitats across the greater Gorongosa-Marromeu landscape, and a sustainable development and human-wildlife coexistence strategy laid the foundations for a strategic program to reintroduce other extirpated carnivore species, particularly those less conducive to natural recovery. The reintroduction program initiated with the translocation of African wild dogs (*Lycaon pictus*) to GNP in 2018.

Wild dogs historically occurred across much of sub-Saharan Africa [26, 27] evolving as an integral component of ecosystems over the past 3.9 million years [28]. Ecologically unique, wild dogs are classified as large carnivores [2, 29], are the only true coursing predator in sub-Saharan Africa [30], are critically important to healthy ecosystems, and fulfill important top-down roles in regulating prey populations [31]. Despite this, wild dog populations have undergone sustained declines over the past half-century driven largely by rapid and severe habitat loss and fragmentation [27, 32], direct persecution by ranchers and pastoralists [27, 33], and as bycatch in a pervasive, illegal bushmeat trade [34]. Currently, wild dogs are restricted to less than seven percent of their historical range [35] and are listed as endangered [27]. It is therefore imperative to strengthen protection of extant populations of wild dogs and their habitats [26], and recover and re-connect populations where feasible. Reintroductions remain an important strategy for the recovery of extirpated populations [16, 36], although a systematic understanding of the successes and challenges of wild dog reintroductions in to open, unfenced ecosystems–which defines most remaining landscapes on the African continent—is lacking in the literature [37].

Determining reintroduction success is complex, needs to account for a spectrum of criteria, and is highly dependent on the temporal scale of the assessment [19, 38]. Short-term success relates foremost to the immediate fate of the release generation and the first wild born generation [37, 39], whereas long-term success relates more to the establishment of a self-sustaining population requiring little or no intervention [40]. Criteria for evaluating reintroduction success should integrate species-specific life-history traits [41] and should clearly define the time scale of the evaluation [19, 38]. For wild dogs, new pack formation is key to demography and population growth [42]; Packs can form both naturally [43, 44] or artificially such as with opposite-sexed and unrelated cohorts being bonded together in pre-release enclosures [45, 46]. Pre-release artificial bonding is a crucial indicator that enhances group cohesion [15, 47], promotes alpha pair establishment [48], positively influences post-release reproduction [47], and is effective in the reintroduction phase for a new population [16, 37]. Post-release survival of founders is also a key indicator [18, 37, 49, 50], as is consecutive years of breeding of the release generation coupled with natural recruitment [37, 39] (Table 1). High rates of breeding and survival and low rates of mortality and emigration underpin population growth and thus over the short-term, population size and growth rates are also important indicators of success [16, 36] (Table 1).

The primary aim of our GNP wild dog reintroduction study was to provide baseline data and an analysis of short-term (first two-years) reintroduction success [37]. We also describe and analyze the effectiveness of artificial group formation in pre-release enclosures and subsequent post-release demography, diet and space use. We evaluate short-term reintroduction success against five categories of indicators (comprised of nine key indicators; Table 1). We hypothesized a successful reintroduction due to high prey availability in GNP [20], low lion density [25], a low level of human impact on wildlife in the park (P. Bouley unpublished data), and a well-coordinated translocation strategy led by wild dog reintroduction specialists. Our final aim was to evaluate which mechanisms could be driving successful reintroduction outcomes for wild dogs in GNP. Considering that lions are the main natural threat to wild dogs [44, 51–53], we predicted that wild dogs would coexist with lions in GNP via spatial partitioning [44, 53–56] and/or dietary niche partitioning [57, 58] and that these behaviors would result

**Table 1. Framework for evaluating short-term success of the wild dog reintroduction to Gorongosa National Park, Mozambique.**

| Category | Indicator [References] |
|---|---|
| 1. Artificial pack formation | 1A. Founder males and females bond as a new pack*, the pack persists as a unit post-release [16, 47] |
| | 1B. Alpha pair established in each pack [48] |
| 2. Natural pack formation | 2A. Dispersal cohorts from founder packs [16, 37, 65, 66] |
| | 2B. Dispersal results in formation of new packs reflecting natural population dynamics [37, 42, 44, 49, 65, 67] |
| 3. Breeding | 3A. Breeding by release generation [37, 39] |
| 4. Survival | 4A. Survival of release generation [16, 18, 37, 49, 50] |
| | 4B. Recruitment of individuals, through raising pups to one year old [18, 39, 49, 66, 68–70] |
| 5. Population growth | 5A. Population size increased [18, 49, 66, 70] |
| | 5B. Positive annual population growth rate [17, 49, 66, 70] |

*only relevant for the first founder pack (Gorongosa pack)

Indicators and their references are taken from multiple species accounts in the in the IUCN Global Reintroduction Perspectives Reports [59–64] as well as from carnivore and/or wild dog specific references.

in wild dog areas of intense use offset to those areas intensively used by lions as well as differences in prey utilization and preference. Humans are also a threat to wild dogs both within and outside of protected areas [33] and we predicted that wild dogs would avoid negative anthropogenic impacts, avoid human settlements and suffer little to no human-induced mortality or injury from snares/traps.

## Materials and methods

### Study site

The Gorongosa-Marromeu landscape located in Central Mozambique (Fig 1) is composed of a highly diverse mosaic of habitats [71, 72]. This landscape is managed as a contiguous 20,000

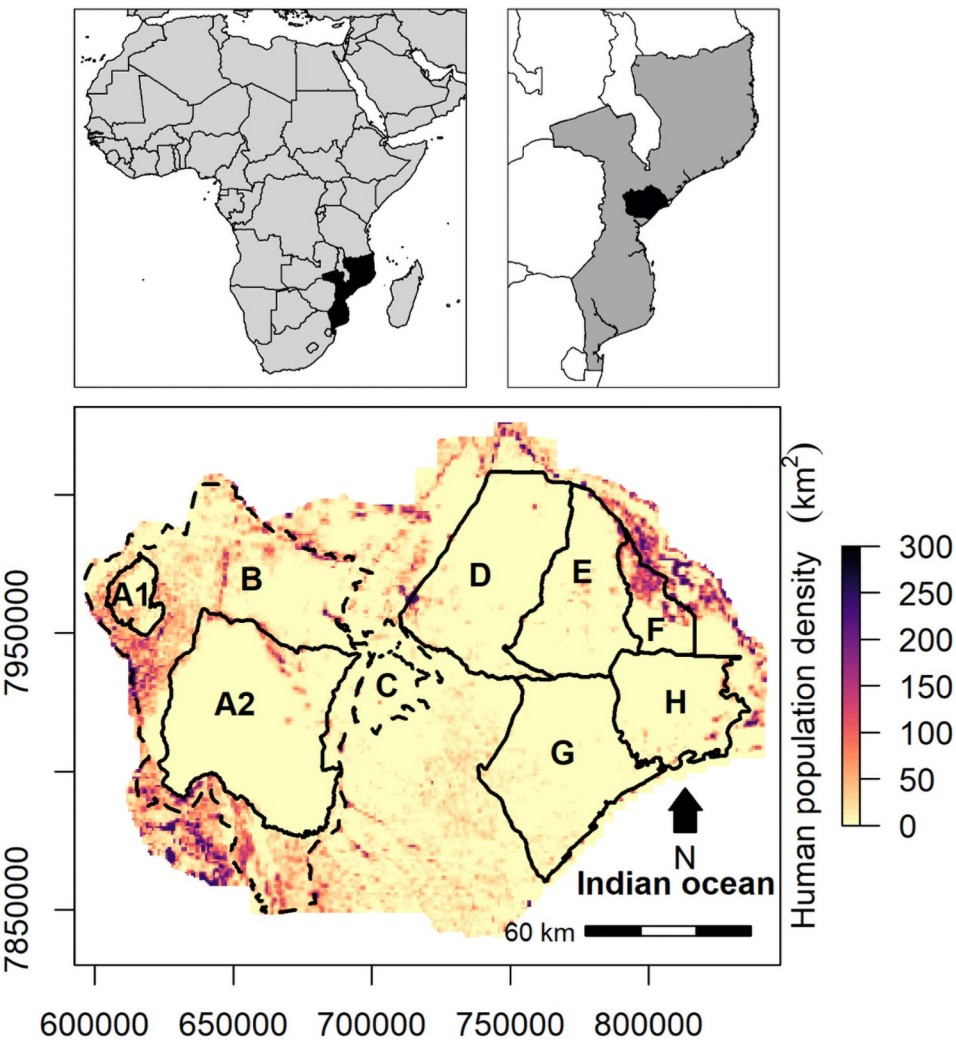

**Fig 1. The location of the Gorongosa-Marromeu landscape (bottom panel) within the Sofala province of central Mozambique (black shading in top right panel) on the east coast of Africa (black shading in top left panel).** In the bottom panel, different protected areas are shown with solid lines indicating official protected area status and dashed lines for buffer-zones and adjacent forestry areas overlaid on human population density (number of people per 1 km²). The human population density data were collected by Gorongosa National Park's Department of Scientific Services. Letters refer to (A1) Mountain section of Gorongosa National Park, (A2) core area of Gorongosa National Park, (B) buffer area of Gorongosa National Park, (C) forestry areas, (D) Coutada 12, (E) Coutada 11, (F) Coutada 14, (H) Marromeu National reserve and (G) Coutada 10.

km$^2$ complex of national park, national reserve, buffer-zones, forest reserves and hunting concessions or "coutadas" (Fig 1). Designated as a potential stronghold for lions [73, 74], the Gorongosa-Marromeu landscape, is also habitat for leopard (*Panthera pardus*), spotted hyenas (*Crocuta crocuta*) and wild dogs which were all present pre-civil war [21] but are currently ephemeral, absent or occur at very low densities.

The unfenced, 4,067 km$^2$ core area of GNP is situated in contiguous landscape of protected areas known as "Gorongosa-Marromeu" (Fig 1; -18.97898˚, 34.35190˚) and positioned at the southern edge of the Great Rift Valley. The park's vegetation types (not including Gorongosa Mountain) can be broadly classified into woodland and floodplain with the 18 km$^2$ Lake Urema at its center. GNP is sub-tropical and characterized by wet summers (November-April) and dry winters (May-October). In 2018, herbivore biomass density in GNP was estimated at ~8,800 kg.km$^{-2}$ or approximately 95% of pre-war levels [75]. Of the 20 focal small- to mega-sized herbivores species present such as oribi (*Ourebia ourebi*), warthog (*Phacochoerus africanus*), bushbuck (*Tragelaphuss ylvaticus*), nyala (*T. angasii*), kudu (*T. strepsiceros*), buffalo (*Syncerus caffer*) and others, composition is currently dominated by a meso-herbivore, the waterbuck (*Kobus ellipsiprymnus*) [20]. Pre- the 1977–92 civil war, a range of carnivores were documented in GNP including *C. Crocuta*, *P. pardus*, *P. leo* and *L. pictus* [21]. Beginning in 2012, GNP launched a long-term, intensive survey of extant carnivore populations. With the exception of ephemeral sightings of a lone spotted hyaena in 2012 and a single male leopard (between 2018 and 2020), an indigenous population of lions is the only extant large carnivore in GNP [25].

## Determining the status of wild dogs in GNP

Wild dogs occurred in GNP prior to the civil war between 1977 and 1992 [21] but were completely extirpated by 1992; declines likely driven by poaching (snaring/trapping), prey depletion, and instability in governance [76]. An intensive survey of large carnivores commencing in 2012 [25] as well as WildCam Gorongosa surveys (https://www.zooniverse.org/projects/zooniverse/wildcam-gorongosa) and the daily tracking of eco-tourism guides in the area, yielded no sightings of wild dogs in GNP between 2012 and 2018. It was concluded that wild dogs were effectively extirpated and natural recolonization from other resident populations in Mozambique was highly unlikely given their status or distance and separation from GNP. While a single, remnant pack of five adults is being closely monitored in a former hunting concession adjacent to GNP, (see area "D" from Fig 1, approximately 70km from the boundary and 130km from the release enclosure in this study), their long-term survival remains precarious given their small pack size. Niassa Reserve, which is home to Mozambique's largest population of wild dogs, is situated north of the Zambezi River (a major and permanent movement barrier) and approximately 600km north of GNP. Limpopo National Park and its partner reserves adjacent to Kruger National Park also contain a wild dog population but are situated approximately 480km south-west of GNP and south of the Savé River. Currently, no formal or informal wildlife corridors are known to exist between Gorongosa and Niassa Reserve or Limpopo National Park or extant populations in Zimbabwe or Zambia, thus making natural recolonization highly unlikely.

## Reintroduction suitability assessment

Considering that the last wild dogs disappeared during or after the civil war and that no recolonizations occurred post-civil war despite extensive effort to protect the landscape, conservation translocations (i.e. reintroductions) were identified as the primary recovery strategy in GNP [26]. To evaluate if GNP was ecologically suitable and whether it was a viable site for a

reintroduction, we implemented two data-driven desktop assessments. Firstly, GNP was among a number of sites assessed in 2015 by the Range Wide Conservation Program for Cheetah and African Wild Dogs (RWCP) undertaken to investigate, identify and update resident, possible, transient, connected, recoverable, extirpated and unknown sites across southern Africa [26].

Secondly, a wild dog feasibility assessment designed by the Wild Dog Advisory Group-South Africa (WAG-SA) to guide reintroductions was used to assess the suitability of potential reintroduction sites. Feasibility assessments for the reintroduction of wild dogs in South Africa have been used for over 20 years with success in recovering wild dog range and population density [36]. The WAG-SA assessment incorporates ecosystem size, preferred prey densities, competitor predator densities, disease history, fencing conditions, potential for human-wildlife conflict and sustainable financial support to support the recovery process into the future (Endangered Wildlife Trust (EWT), unpublished data).

Through these two desktop assessments, GNP was identified as 'recoverable' area by the RWCP and a conservation translocation was considered the best option of recovering the population given the unlikelihood of reconnecting GNP to any other confirmed resident populations nationally or in adjoining countries [26]. GNP also scored 76 (out of a possible 100) in the WAG-SA feasibility assessment (EWT, unpublished data). As GNP is not fenced, the potential for exposure to disease at the human-wildlife interface was where GNP was deemed most vulnerable. However, with an average score of 68 among other protected areas in South Africa (EWT, unpublished data) conditions in GNP suggested a reintroduction was feasible. GNP supports abundant herbivore populations [20] and based on the herbivore biomass in GNP, the carrying capacity model [77] estimated that GNP could sustain a minimum of 82 wild dogs or approximately eight packs (assuming average pack size of 10 individuals). GNP also has a full-time, locally staffed wildlife management team that includes rangers, veterinarians, carnivore specialists and data analysts who rigorously track and monitor wildlife resources and relevant anthropogenic factors, e.g. snaring pressure and human-wildlife conflict. Strategic deployment of professionally trained rangers across the landscape also resulted in a 65% reduction in snaring pressure with poaching of lions subsequently declining by 95% between 2015 and 2018 (Dr. Rui Branco, pers comm). GNP managers were therefore confident that threats that lead to the historical extirpation of wild dogs in GNP had been alleviated. Overall, the feasibility assessments suggested that GNP could support the recovery of wild dogs and a conservation management decision was made to implement a coordinated reintroduction of wild dogs into GNP.

## Sourcing of founder packs

Due to the success of the metapopulation in South Africa [36], multiple wild dogs were available as surplus individuals and packs. Considering this availability and their high levels of genetic diversity [78], wild dogs sourced from the metapopulation in South Africa were determined to be appropriate founders for the reintroduction. Across Africa, there are two main genetic clades that are distinguished for wild dogs referred to as the E genotype from eastern Africa and the S genotype from southern Africa [79]. Their composition is suspected to reflect a scenario of extinction of eastern African populations and subsequent recolonization from western or central Africa, with secondary migrations between eastern and southern Africa [35]. The E and S genotypes have close evolutionary relationships and are not representative of subspecies or evolutionary significant units. Large admixture zones exist between the two clades in Botswana, Zimbabwe, and south-east Tanzania indicating that large-distance dispersal (>400 km) has occurred until recently. Although the wild dog population in

Mozambique has never been studied thoroughly, the demographic history suggests that it likely lays within a hybrid zone between the E and S clades and that reintroduction of S geno-types would be appropriate for GNP. The decision to use wild dogs from South Africa was therefore not expected to have any deleterious effects on the genetics of any extant population close to GNP.

We conducted the reintroduction in two phases, each comprising the reintroduction of a single pack. We initiated phase one in 2018 which involved the translocation of nine adult males from the uMkhuze section of iSimangaliso Wetland Park in KwaZulu-Natal (-27.64026˚, 32.15859˚) and one adult and five yearling females from a free roaming pack in the Magudu area in KwaZulu-Natal (-27.50140˚, 31.55088˚). These male and female groups were unrelated [78] and were deemed ideal for artificial bonding to create a new pack in the pre-release enclosure.

We initiated phase two in 2019 which involved the translocation of a recently naturally formed pack of nine adult males and six adult females from Khamab Kalahari Game Reserve (-27.33249˚, 31.84555˚). The reserve had reached their carrying capacity for predators and a management decision was made to remove the pack. This pack was genetically distinct from the 2018 pack and comprised of lineages from Zimanga Game Reserve in KwaZulu-Natal (-27.5741˚, 32.0030˚), Madikwe Game Reserve in the North West (-24.7604˚, 26.2777˚) and from individuals that naturally immigrated into Khamab, assumed to have originated from south-west Botswana.

## Translocations

Wild dog reintroductions are more successful with soft release from holding enclosures than hard releases [15, 18, 37]. Moreover, when artificially forming new packs, unrelated groups of males and females need to undergo a period of pre-release bonding to one another to enhance pack formation and increase short-term and long-term success [47]. Using small (~5,000m$^2$) and electrified holding areas and an olfactory acclimatization technique [48], bonding between unrelated opposite sex groups can be enhanced. Thus, we constructed a permanent single compartment 5,000 m$^2$ enclosure with ample shade-cover, a shallow swimming area and water trough to receive, hold, acclimatize and socially integrate groups prior to their soft-release into GNP. The enclosure was constructed in the core area of GNP (area "A2" in Fig 1) with ample distance from any boundaries or community areas and included robust electric fencing to keep the pack inside the fence safe from predators such as lion, and to prevent individuals from escaping the enclosure prematurely. The design of the enclosure minimized exposure to people while permitting appropriate access by vehicle for visual monitoring and for routine feeding, cleaning and water re-supply.

For each translocation, we compiled individual photographic identification kits that comprised information on each individual's area of origin, age, current pack and status, collar type, unique identification number and photographs of the left- and right-side coat patterns and distinguishing marks. This allowed all subsequent observations to be related to unique individuals. Additionally, for both phases, we administered rabies and canine distemper virus (CDV) vaccinations to all wild dogs while they were sedated during air-travel.

**Phase one—2018 (Gorongosa pack).** Six females (one adult and five yearlings) and nine adult males were captured and transported on the 29$^{th}$ of March 2018 to adjacent compartments of an enclosure at Phongola Nature Reserve, South Africa (-27.33291˚, 31.84462˚). We deliberately placed the unrelated and opposite sex groups adjacent to one another to allow for a controlled group integration [45, 46]. On the 16$^{th}$ of April 2018, we sedated all dogs using a combination of Zoletil and Medetomidine and translocated the two groups via air-charter to

GNP, a distance of ~1,000km and a journey of 3.5 hours. We collared every wild dog, specifically fitting two satellite collars, one on the oldest male who was presumed to assume alpha status, and one on the alpha female. An additional 12 VHF collars were fitted on the remaining females and males. To accelerate the bonding process, we physically rubbed sedated males and females together to exchange individual scents and build familiarity between the two sex groups with the goal of reducing aggression on wake up and to help promote pack cohesion [48]. Unfortunately, a single adult male died on arrival at the enclosure, cause unknown. This pack was named "Gorongosa."

**Phase two—2019 (Pwadzi pack).**    We captured and translocated a pack of nine adult males and six adult females from Khamab Kalahari Reserve on the 5th of November 2019. The dogs were fully sedated using a combination of Zoletil and Medetomidine and flew the two groups together a distance of ~1,300km and a journey of 4.5 hours. One arrival at the airfield, the pack was driven 8-km to the enclosure. The presumed alpha male and alpha female were both fitted with satellite collars and an additional 12 VHF collars were distributed among the remaining individuals. This pack was named "Pwadzi."

## Captive bonding, monitoring and release

The acclimatization period during which the packs were captive in the enclosure was crucial to facilitate bonding (for phase one) and to allow both packs reintroduced time to adjust to their new surroundings. An individual life history database and identification kit was compiled for each pack identifying each dog with a unique number and with photographs of their coat-patterns and other identifying characteristics; this facilitated accurate and detailed observations of behavior and associations. We implemented a daily schedule of monitoring and maintenance procedures at the enclosure. We monitored wild dogs twice daily; at first light for two hours and again for two hours prior to sunset, corresponding to their crepuscular periods of activity [45, 46, 48]. During each monitoring session, we recorded the number of individuals in the holding enclosure, group association behavior, aggressive interactions, mating events and hoo-calling while performing maintenance checks at enclosure to reduce the likelihood of escape. We fed the wild dogs every 2–3 days with freshly culled, whole waterbuck and replaced water in the trough every 1–3 days. No wild dogs escaped from the enclosure during this period. From the behavioral monitoring sessions at the enclosure, we estimated pack cohesion and acclimatization in order to identify suitable timing of release and were guided by reintroduction time frames that optimize short-term and long-term success [47]. In the week prior to release, we administered booster vaccinations for CDV via drop-out darts. Packs were passively released from the enclosure via opening the main gate with observers placed at a distance of 100 m from the gate. The first pack (Gorongosa pack) was released on the 15th of June 2018 after a period of eight weeks in their enclosure, while we released the second pack (Pwadzi pack) on the 10th of December 2018 after a period of five weeks in the enclosure. Due to the natural pack formation of Pwadzi pack at Khamab, they were kept in the enclosure for a shorter period than the Gorongosa pack.

## Wild dog data collection post-release

We collected and utilized data from our intensive post-release monitoring that covered a total of 28 consecutive months spanning from June 2018 to September 2020. This represents a short-term that covered three consecutive denning seasons and allowed packs enough time to attempt to recruit individuals through reproduction and pup survival. It also allowed enough time for packs to settle into their new area and form territories.

**Post-release monitoring.**    While we attempted to locate the packs multiple times per week, this was reliant on accessibility, mostly constrained by weather conditions. We used the GPS and VHF collars (see Collars and GPS data section below) to locate wild dogs. Once located, wild dogs were observed from a 4x4 vehicle with 1–2 observers recording data. At each sighting, we recorded (1) number of individuals, identities, sex and age of individuals, (2) health condition of individuals and any injuries, (3) dominance (known from overmarking, mate-guarding and scent marking), (4) pregnancies, (5) hunting success, and (7) any behavioural interactions with conspecifics and lions. We also relied on opportunistic field ranger and tourist reports to supplement our dataset. As each wild dog is identifiable from unique coat patterns [80], we utilized photographs and videos from sightings to confirm individuals present. We verified all identities against the photographic identification kits constructed during translocation. At the worst resolution, each individual was seen and recorded at least once per month. Consequently, we built individual and group-level life histories for wild dogs in GNP at a monthly resolution from June 2018 until 30 September 2020. We then used the demographic data to assess monthly (1) group composition, (2) group size, (3) reproduction and recruitment, (4) inter-group movement dynamics, (5) survival, (6) mortality, (7) GPS location and (8) kills.

**Breeding.**    We used GPS collar data and direct observations to determine the timing and location of denning. Birth females were known from monitoring prior to denning based on direct observations of alpha pairing, mating, visible signs of pregnancy, and females not joining pack in hunting events. Den sites were initially confirmed by cluster patterns observed from satellite-collared females and the pattern of packs returning to the same site twice a day over consecutive days/weeks. We also recorded the coordinates of dens and the week of commencement of denning. We estimated litter size from camera traps placed at identified den sites, recording both images and videos of pack structure and behaviour. We rotated cameras on a weekly to biweekly schedule as access and batteries permitted. Disturbance and stress to packs was minimized by replenishing cameras during periods of the day when packs were out hunting, usually in the morning. Camera trap data allowed us to add photographs pups (i.e. non-founders) into our individual life history database, assign age and sex, and track individual survival of pups from emergence. We generally recorded pups emerging from dens approximately 3–4 weeks after females went underground. This is the typical age when pups first emerge from underground [44] and we are confident that this is comparable to litter sizes from other wild dog studies [66, 81–84]. We also counted the number of surviving pups o three, six, and 12 months as important age milestones to determine the proportion of pups raised and recruitment. We defined recruitment as the number of pups raised to 12 months old (i.e. yearling) as a proportion of the population size (adults and yearlings). The proportion of pups surviving to six, and 12 months does not include the six 2020 litters because our short-term evaluation ended prior to these pups turning six and 12 months old.

**Population size and growth, survival and mortality.**    We defined three age class categories for wild dogs including pup (< one year old), yearling (1–2 years old) and adult (> 2 years old). We defined a pack as a group of wild dogs with at least one adult male and one adult female, dispersal group as a cohort of a single-sex individuals and splits as a mixed sex cohort that permanently left their most recent pack. We generally classified pack and population size as the sum of adults and yearlings unless otherwise stated. We used two measures for population density: 1) the population size per 100 km$^2$ and 2) as the number of packs per 100 km$^2$ to allow for comparisons with other populations, and because the pack is the unit of reproduction. We also estimated the annual instantaneous population growth rate as the natural log of the current year's population size divided by the previous year's population size [85], as well as population growth rate for adults, yearlings and pups combined. Through the field data

collection, we assigned alive or dead to each individual at the end of every month. From these monthly data of individual fate, we were able to estimate survival as the number of individuals alive at the end of the study relative to the total number of individuals recorded during the 28 months. This method included both founders and non-founders (i.e. pups). Any individual observed to have died and the carcass or collar recovered was assigned as a confirmed mortality event where we also determined the month of mortality and the cause as human (road accident, indirect snaring, direct shooting), natural (predation, intra-specific, hunting) or disease (rabies, distemper) [33]. If an individual was not confirmed to have died and was not alive at the end of the study, we assigned it as disappeared and estimated disappearance month as one month after last confirmed observation. The sex and age class of dead and disappeared individuals were also recorded. We also recorded changes in group association as evidenced by successful dispersal (i.e. natural pack formation with opposite sex group), unsuccessful dispersal (have yet to form new pack) and splits.

**Diet and prey preference.** We recorded all observed wild dog kills, or carcasses located that were confirmed to have been predated by wild dogs. At each kill or carcass site, we recorded the date, location, prey species, prey sex and age class (where possible). We utilized the GNP biennial herbivore census data [75] to determine the amount of available prey biomass for wild dogs. Here, we defined available wild dog prey as those species that fall within a defined weight range as their preferred prey [86] and any additional species that we observed wild dogs having successfully predated upon (e.g. waterbuck). We applied Jacob's Index of selection [87] to assess prey selection using the number of observed kills and available prey biomass. We separated waterbuck biomass into two distinct categories related to the large difference in their biomass relative to each habitat type [75]: (i) grassland/woodland waterbuck and (ii) floodplain waterbuck. We also assessed lion kill data between 2012 and 2020 (P. Bouley unpublished data) using Jacob's Index of selection thereby allowing us to determine if wild dogs were employing dietary niche partitioning as a lion coexistence strategy.

**Collars and GPS data.** We fitted 42 collars on 30 unique wild dogs (only adult-sized individuals) during this study. Translocated individuals were fitted with collars by a veterinarian during the transport phase, and post-release individuals were sedated and collared by veterinarians free-darting from a 4x4 vehicle at a range of 20-35m and using a partially reversible combination of Medetomidine (0.4mg) and Ketamine (45mg). Overall, we fitted 28 satellite collars (24x African Wildlife Tracking; 2x Sirtrack Iridium; 2x Wildtrack) and 14 VHF collars (6x African Wildlife Tracking; 8x Animal TrackEm). Satellite collars weighed 450-500g and were within 2% of body weight. Multiple individuals per pack were collared to ensure robust data in the event of collar failing. This allowed us to rapidly relocate a pack, but also to more accurately track dispersal and mortalities when they occurred. For spatial analyses, we only used GPS data from one individual per pack at any time to reduce pseudo-replication. We also attempted not to select collar data from breeding females to avoid small and non-representative territories that occur during denning as breeding females tend to stay near the den for most of the denning season [44]. Consequently, we included GPS data from nine of the 30 collared individuals (S1 Table). Satellite collar fix schedules varied but recorded six fixes per day on average. The full dataset contained 7,944 GPS fixes and was truncated to include a maximum of four fixes per group per day, representing two resting periods and two movement periods, each separated by at least six hours. We considered these temporal intervals adequate to avoid spatio-temporal auto-correlation between consecutive locations without losing relevant information on the ecology of the animals [44]. The truncated dataset thus included 3,048 fixes with a mean number of 339 fixes per group (S1 Table).

**Lion spatial data.** Lions pose the largest natural threat to wild dogs through direct interference competition [33, 44, 52, 88] that can limit wild dog populations [44, 53] or even

contribute to local extirpation [57, 89]. GNP has a recovering and growing population of lions [25]. To estimate lion spatial extent in GNP, we used lion satellite collar data collected by the GNP's Conservation Department between June 2018 and September 2020. We used GPS data from 14 groups of lions (five prides, four male coalitions and five male dispersal groups), all fitted with African Wildlife Tracking satellite collars (S1 Table). Lion collars were programmed to record fixes every 2–4 hours and we truncated the lion GPS dataset in the same way we did for the wild dogs, reducing pseudo-replication, and sampling the same time periods as for wild dogs.

## Territory estimation

**Kernel Utilisation Distributions and territory size.** We used the truncated spatial dataset for wild dogs and lions (S1 Table) to estimate territories using Kernel Utilisation Distributions (KUDs). We defined KUDs with a pixel size of 1 km$^2$ based on bivariate kernels using the *kernelUD* function from *adehabitatHR* package in R. We estimated our smoothing parameter as $h = 2,100$ based on other wild dog studies that selected $h = 2,400$ m [90, 91] for GPS data [92]. We used this same smoothing parameter value for the lion KUDs. For wild dogs, we estimated group-level KUDs for the first 14 months (June 2018-July 2019), for the second 14 months (August 2019-Sepetmeber 2020) and for the entire 28-month study period. We used the *getverticeshr* function in *adehabitatHR* package in R and specified the WGS84 geographic projection to extract the area size (km$^2$) enclosed by the 95% and 50% isopleths extracted from KUDs for each half of the study and for the entire study period for wild dogs. For lions, we only created KUDs for the entire study period and then extracted the 95% and 50% isopleths. We defined the total territory area for a group by the 95% isopleth while the core territory area was delineated by the 50% isopleth.

**Overlap area.** We assessed spatial interactions between neighbouring wild dog group dyads using overlap as an index for territoriality [93, 94]. We estimated overlap area (km$^2$) between a pack's range for each neighbouring dyadic association to determine what proportion of each pack and dispersal group's territory overlapped with each of its neighbours. We estimated this for both the 95% and 50% isopleths extracted from the KUDs created over the whole study period. We used the merged 95% and 50% isopleths for all wild dogs and the merged 95% and 50% isopleths for all lions to estimate the amount of overlap between these two carnivores at the level of the population. We did this by relating the proportion of overlap in area size between the 95% and 50% wild dog territories to that of the 95% and 50% lion territories for the entire 28-month study period.

**Habitat use.** We determined the proportion of fixes per wild dog and lion group from the truncated dataset that fell within each of the two broad vegetation categories (woodland or floodplain). We were thus able to contrast species-specific habitat use in GNP corresponding to second order of selection [95]. We excluded fixes that fell outside the official protected area boundary of the park for the habitat use analysis. We also determined the number of wild dog dens in each of the two broad vegetation types. Some female wild dogs denned in the same site as other females in the same pack within the same year (S2 Table) and we did not include these duplicates in this analysis (n = 23 unique den locations).

**Range expansion.** To determine if the wild dog population increased its range over time, we merged the extent of the group-level 95% isopleths for the first 14 months of the study period and for the second 14 months of the study period. We also estimated the proportion of the park used by wild dogs using the merged 95% isopleths for each group covering the entire study period to help guide managers with decisions on potential future introductions. We also developed monthly 95% minimum convex polygons (MCPs) for packs using the *mcp* function

from the *adehabitatHR* package v0.4.15 [96]. We did this to represent the total extent of the area covered by wild dogs without make inferences about use.

## Ethics and permit numbers

All field work was conducted in accordance with the American Society of Mammalogists guidelines [97] with collars weighing well below 5% of animal body weight. The fieldwork was approved by the University of Pretoria's Ethics Committee (EC018-18) and by Gorongosa National Park under research permit number PNG/DSCi/C108/2018. The 2018 translocation was permitted under import permits that included a Mozambican Conservation certificate (no. 0004/2018) and a Mozambique State Veterinary certificate (no. 02AB/DEV/2018) as well as exports permits from South Africa including an Ezemvelo Kwazulu Natal Wildlife permit (no. OP 1544/2018) and South Africa State Veterinary permit (no. 045312). Similarly, in 2019 the translocation was permitted under a Mozambican Conservation certificate (no. 00265/2019), a Mozambican State Veterinary certificate (no. 27AB/DEV/2018), a South African North West Province Permit (no. NW 14879/10/2019) and South African State Veterinary permit (no. 018535). Consistent across all permits were that the immobilization and transport of all wild dogs was conducted by a qualified and licensed wildlife veterinarian. Additionally, we ensured as minimal stress to the wild dogs as possible by ensuring all individuals were blindfolded during the whole translocation phase with a veterinarian and an additional technical expert continuously monitored heart rate, breathing, body temperature and degree of sedation.

## Statistical analyses

While we generally provide descriptive statistics in our study, we performed parametric (i.e. linear models) and non-parametric (i.e. Chi-squared) tests to evaluate relevant relationships. We used linear regression models to test (1) the effect of year (2018, 2019 and 2020) on population size (adults, yearlings and pups combined), (2) the effect of year (2018, 2019 and 2020) on the number of packs, and (3) the effect of month-year on the MCP size for all dogs combined (to see if there was an asymptote or not). We constructed and evaluated these models using the *lm* function in RStudio.

We used Chi-squared tests to investigate (1) the effect of month on the number of litters observed, (2) the proportion of pups surviving to 3, 6, and 12 months and (3) the effect of group type (founder vs non-founder) on the proportion of animals alive at the end of the study. We also used Chi-squared tests to investigate the effect of vegetation type (woodland versus floodplain) on (4) the number of wild dog GPS fixes in each vegetation type, (5) number of wild dog kills in each vegetation type and (6) the number of lion GPS fixes in each vegetation type. We then used Chi-squared tests to see determine if there were differences in wild dog versus lion use of (7) woodland vegetation and (8) floodplain vegetation and (9) the effect of vegetation type (woodland versus floodplain) on den site location.

We performed all statistical analyses and constructed all figures in the RStudio GUI, desktop version 1.1.456 [98] that used R version 3.5.1 [99] for Windows, using functions in the *stats v3.5.1* package [98], *ggplot2 v3.1.0* package [98] and we plotted maps using the *plot* function from *graphics* base package [98].

## Results

### Phase one—Gorongosa pack

We observed the adult female (Beira) mating with one of the older male's (Mutondo) within two weeks of being together in the enclosure (i.e. at the end of April 2018). We observed the

close association of Beira and Mutondo with consistent overmarking between the two and them resting together. This indicated the formation of an alpha pair (indicator 1B, Table 1). These were the two oldest individuals in the enclosure. Beira and Mutondo continued mating into the first 10 days of May 2018. Two younger males (Mussapassua and Mini-mini) were observed mating with Beira on the 9th, 10th and 12th-14th May. One large fight ensued on 11 May 2018 between the older and younger groups of males and resulted in facial scars on one younger male. Little aggression was observed thereafter and the eight males continued feeding and resting together with the females in the enclosure. It was confirmed during the week pre-ceding release that Beira was visibly pregnant. After their release on 15 June 2018, the eight males and six females remained together (indicator 1A, Table 1). Only 10 months after this a subset of individuals (the three older males and one subordinate, adult female) split to form a new pack (see Natural pack formation section below). Four females (including Beira, the alpha female) and the five younger males of the original founders remained together in the original Gorongosa pack until further natural dispersals of subordinate females took place in August and December 2019.

### Phase two—Pwadzi pack

We did not observe signs of aggression during their entire period in the enclosure and we observed close association between one female (Matenga) and one male (Nhamaguena) that suggested the formation of an alpha pair (indicator 1B, Table 1). We did not observe mating during this group's enclosure period. In contrast to the Gorongosa pack, post-release the 15 individuals only stayed together for four days before the first of two dispersal events occurred (see Natural pack formation section below). Despite the dispersals, by the end of this study period, one female (not the original alpha female, who died a natural death in April 2020) and seven males of the original founders have remained together in the Pwadzi pack (indicator 1A, Table 1).

### Natural pack formation

We documented 12 of the 29 founders change their group association (Fig 2; S3 Table). This was in the form of dispersals (n = 5 events) and a split (n = 1 event) from the original, artifi-cially formed Gorongosa pack and the Pwadzi pack that resulted in the formation of two new, naturally formed packs (Fig 2; S3 Table) (indicators 2A and 2B, Table 1). We observed one subordinate female from the Pwadzi pack leave the pack when she started denning in May 2020 (Fig 2; S2 and S3 Tables). While this is technically a dispersal, this female is currently accompanied only by her remaining pups and it remains to be seen if she rejoins Pwadzi pack, joins another pack or remains with her pups.

### Breeding

We recorded a total of 11 pregnancies across the 2018, 2019 and 2020 denning seasons but we only documented successful litters emerging in the 2019 and 2020 seasons (indicator 3A, Table 1). Of the 11 pregnancies we only observed nine litters; for the two unconfirmed litters, the pups never emerged from their dens. The single 2018 litter of the original founding Goron-gosa pack was unsuccessful and was abandoned shortly after denning, the cause attributed to predation by an African rock python (*Python sebae*). The second unconfirmed litter was aban-doned by one of the two beta-females of the Gorongosa pack in 2020 (S2 Table). We observed 82 pups born (i.e. non-founders) over two consecutive breeding seasons in 2019 and 2020 (indicator 3A, Table 1). Mean litter size for the nine confirmed litters was 9.11 ± SE1.06 (range 5–15). Birth was highly seasonal across years with all litters emerging between May and July

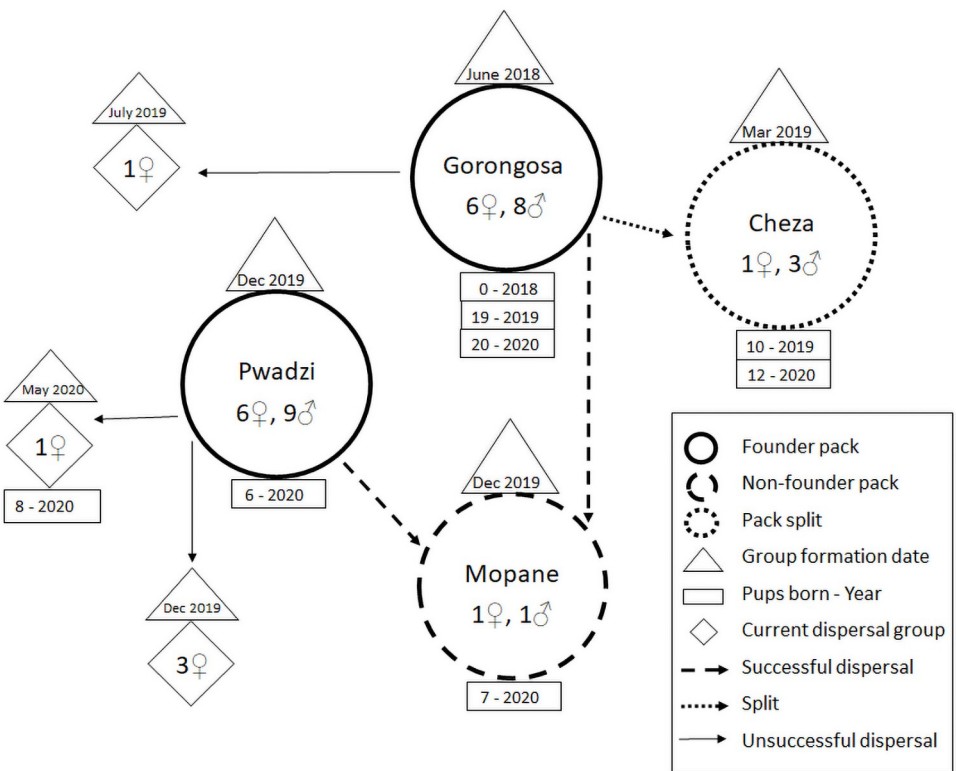

**Fig 2. Group association dynamics of the reintroduced wild dog founders in Gorongosa National Park between June 2018 and September 2020.** The reintroduction of two founder packs (Gorongosa and Pwadzi) resulted in two further pack formations, two successful dispersals and three ongoing dispersal events (one of these is a female with pups she had after dispersing). The number of males and females in Gorongosa and Pwadzi packs show the pack size at the time of release.

($\chi^2_{(11)}$ = 51.07, p < 0.001). Multiple maternities per pack was standard for this population with seven of the nine confirmed litters being born when more than one female per pack gave birth (S2 Table).

## Survival and mortality

Of the 82 pups born, 62 to survived to three months of age (76% survival), 23 survived to six months of age (28% survival), and 23 survived to one year old (28% survival) (indicator 4B, Table 1). Only considering the litters born in 2019 (n = 29 pups), 26 survived to three months (90% survival) and 23 survived to six and twelve months (79% survival). There is no discernible difference in pup survival relative to the three-, six- and twelve-month age milestones ($\chi^2_{(2)}$ = 0.03, p = 0.99) and the population had successfully recruited 23 individuals by September 2020, giving a recruitment rate of 0.47 (indicator 4B, Table 1).

Of the 111 individuals marked and counted during the study, 81 were still alive at the end of the 28 months resulting in a survival of 73% (indicator 4A, Table 1). When comparing survival between founders and non-founders, there was a higher survival rate for founders (25 of 29 individuals, i.e. 86% survival–indicator 4A, Table 1) than non-founders (56 of 82 individuals, i.e. 68% survival; $\chi^2_{(1)}$ = 4.02, p = 0.04).

We observed 27%, or 30 of the 111 individuals, to have died or disappeared from the population. The majority of these were pups (90%) and it is likely that these were all mortalities although only one pup carcass was visually confirmed outside a den. Of the three confirmed

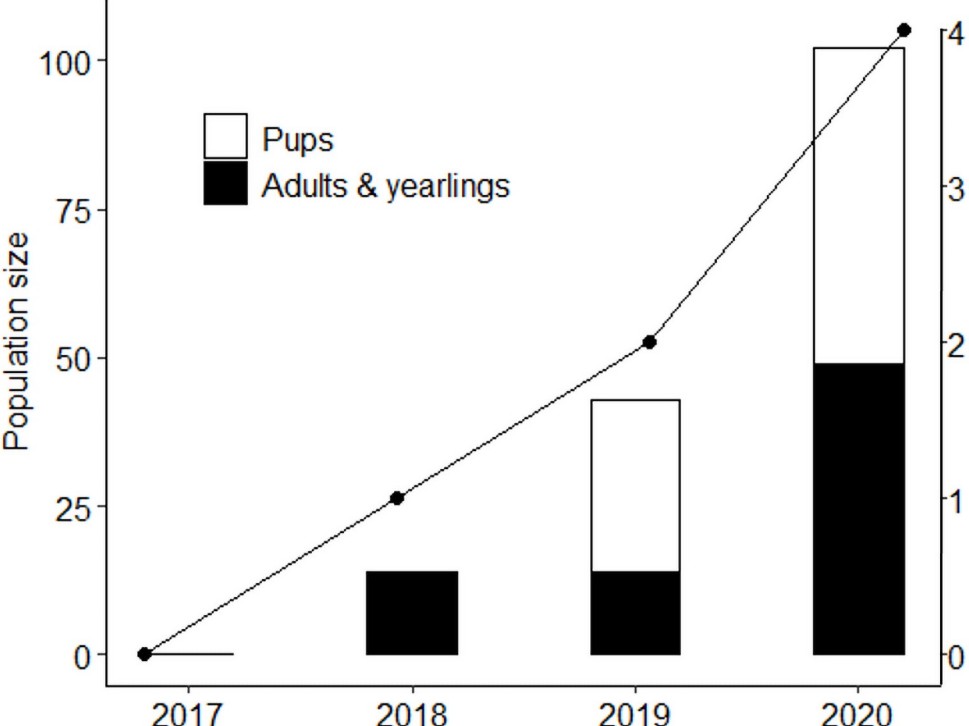

**Fig 3. Annual time series of the wild dog population in Gorongosa National Park from one year prior to their reintroduction (in 2017) until 2020.** Counts were taken in June of each year representing the midpoint in the annual breeding season. Bars represent the total number of individuals in the population (primary y-axis) with the solid line representing the number of packs (secondary y-axis).

adult mortalities we documented, one male was killed by lions, a second suspected killed by lions (based on post-mortem and observed signs of trauma on front leg-bones, although not 100% conclusive given the time that had elapsed) and one female was found dead in a collapsed den. One additional male disappeared during the study and we could not confirm if he had died or dispersed. Overall, we observed no human-induced mortalities in this wild dog population.

## Population size and growth

Over the 28-month study period, we identified 111 unique individuals in the population (29 founders, and 82 non-founders including pups). Population size peaked in June 2020 at 102 individuals (49 adults and yearlings, 53 pups) across four packs and three dispersal groups (Fig 3, Table 2). The wild dog population size in GNP has increased significantly since their reintroduction in June 2018 (adult, yearlings and pup linear model: $F_{(1,2)} = 33.50$, p = 0.04, $R^2 = 0.92$; number of packs linear model: $F_{(1,2)} = 1.30$, p = 0.02, $R^2 = 0.97$; Fig 3) (indicator 5A, Table 1). By the end of September 2020, there were 81 individuals (48 adults and yearlings, 33 pups) across four packs and three dispersal groups. Population growth rate (for adults, yearlings and pups combined) was positive (*r* = 1.12 in 2019, *r* = 0.86 in 2020; indicator 5B, Table 1).

## Diet and prey selection

We observed 102 wild dog kills comprising nine prey species. The majority of kills comprised bushbuck and waterbuck (Fig 4b, S4 Table) but wild dogs only selected for bushbuck (Fig 4a,

**Table 2. Descriptive patterns of annual population dynamics for the reintroduced wild dog population in Gorongosa National Park between June 2018 and September 2020.**

| Variable | Mean ± SE | Range |
|---|---|---|
| AY count | 25.67 ± 11.67 | 14–49 |
| Pup count | 27.33 ± 15.32 | 0–53 |
| AYP count | 53.00 ± 25.89 | 14–102 |
| Population density (AY/100km$^2$) | 0.69 ± 0.31 | 0.38–1.32 |
| Population density (AYP/100km$^2$) | 1.42 ± 0.70 | 0.38–2.74 |
| Pack size (AY) | 10.04 ± 0.63 | 6.5–14 |
| Pack size (AYP) | 15.88 ± 0.86 | 7–25.5 |
| Number of packs | 2.33 ± 0.88 | 1–4 |
| Pack density (/100km$^2$) | 0.06 ± 0.02 | 0.03–0.11 |
| Number dispersal groups | 1.00 ± 1.00 | 0–3 |

AY = adults & yearlings combined, AYP = adults, yearlings & pups combined.

S4 Table). We detected differences in diet and prey selection between wild dogs and lions regarding species and age class (Fig 4, S4 Table). For example, lions did not prey on bushbuck whereas bushbuck makes up 37% of wild dog diet (Fig 4b, S4 Table) and are preferred prey (Fig 4a). Additionally, wild dogs rarely preyed on warthog (the exception being piglets, Fig 4b, S4 Table), while adult and subadult warthog comprised 63% of observed lion diet (Fig 4b, S4 Table). Of the total adult waterbuck predated by each species, 81% of those taken by lion were male and 15% female (n = 26) compared to wild dogs who killed male and female waterbuck equally (S4 Table). Of the 23 confirmed waterbuck kills made by wild dogs, 10% occurred on the floodplain (versus woodland habitats), compared to 67% of the 27 confirmed waterbuck kills made by lions (S2 Fig).

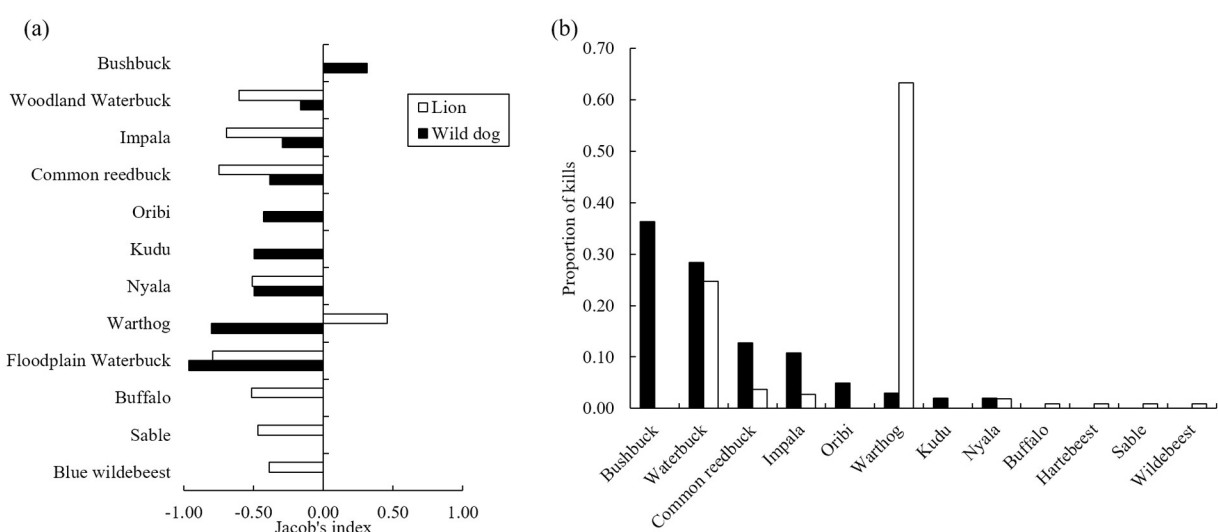

**Fig 4. Comparison between wild dogs (n = 102 kills) and lions (n = 108 kills) in Gorongosa National Park between June 2018 and September 2020 in relation to (a) Jacob's electivity index and (b) the proportion of each prey in the diet.**

## Territory size and overlap

Wild dogs in GNP formed distinct territories, concentrating mainly in the western part of the park and appeared to avoid high human density areas (Fig 5). Wild dogs incorporated the release enclosure as part of their territories, but all groups eventually established core territories that excluded the enclosure, with the exception of the first pack released in 2018 (Fig 5). Total territory size for the four packs was $356 \pm 62$ km$^2$ (mean $\pm$ SE) and core territory size was $71 \pm 14$ km$^2$ (mean $\pm$ SE). Total areas used for the Mucodza and Xivulo dispersal groups was $613 \pm 92$ km$^2$ (mean $\pm$ SE) and their core territories were $118 \pm 12$ km$^2$ (mean $\pm$ SE). The mean ratio of core to total territory size was $0.20 \pm 0.01$ km (mean $\pm$ SE, range 0.18–0.22) for all six groups of wild dogs in GNP (Fig 5). Wild dogs in GNP overlapped with conspecifics and thus did not maintain exclusive territories. The mean proportion of overlap between pack territories was $0.21 \pm 0.02$ km$^2$ (mean $\pm$ SE) while overlap in core territories was $0.07 \pm 0.03$ km$^2$ (mean $\pm$ SE). The two dispersal groups overlap by 48% in the total territory area and 22% in the core.

## Habitat use

Wild dogs used vegetation types differently in the park, with significantly higher use of woodland compared to floodplain areas ($\chi^2_{(1)} = 2922.6$, $p < 0.001$; Fig 6; S1 Fig). Additionally, wild dogs made significantly more kills in woodland (n = 63 kills) than floodplain (n = 14 kills) areas ($\chi^2_{(1)} = 59.84$, $p < 0.001$; S1 Fig). Lions exhibit the same pattern of habitat use, where they used woodland areas more than floodplain areas ($\chi^2_{(1)} = 39.37$, $p < 0.001$; Fig 6). However, wild dogs used woodland areas significantly more than lions did ($\chi^2_{(1)} = 91.75$, $p < 0.001$; Fig 6) while using floodplain areas significantly less than lions did ($\chi^2_{(1)} = 759.3$, $p < 0.001$; Fig 6). Broad vegetation type affected den site location, with significantly more den sites in woodland (n = 28) than floodplain (n = 1) ($\chi^2_{(1)} = 34.78$, $p < 0.001$; S2 Fig). Additionally, all breeding dens were located in core areas for the four packs, except one den for Cheza just outside of core (S2 Fig) and we note interesting den site observations in S1 Appendix.

## Avoidance of lion territories

Wild dog territories in GNP overlapped with those of collared lion prides (n = 5), coalitions (n = 4) and dispersers (n = 5) (Fig 7). Overlap between wild dogs and lions was higher in the total territory areas (56% overlap in the 95% isopleth; Fig 7A) while core territory areas were more exclusive between wild dogs and lions (17% overlap in the 50% isopleths; Fig 7B). Collared lions accounted for approximately 10% of the population and 68% of the extent of area used by lions in the park (P. Bouley, unpublished data) and we acknowledge there are uncollared individuals/groups during the period of this study.

## Range expansion

The population expanded their range over two-fold (117% increase) from the first half of the study period (501km$^2$; 13% of park; June 2018 –July 2019) to the second half (1,089km$^2$; 29% of park; Aug 2019 –Sep 2020) (S3 Fig). Over the total 28-month study period, wild dogs in GNP used 32% of the park. The total monthly area used by packs in the park has increased over the 28-month study period ($F_{(1,26)} = 75.11$, $p < 0.001$, $R^2 = 0.74$) and has yet to reach asymptote (S4 Fig).

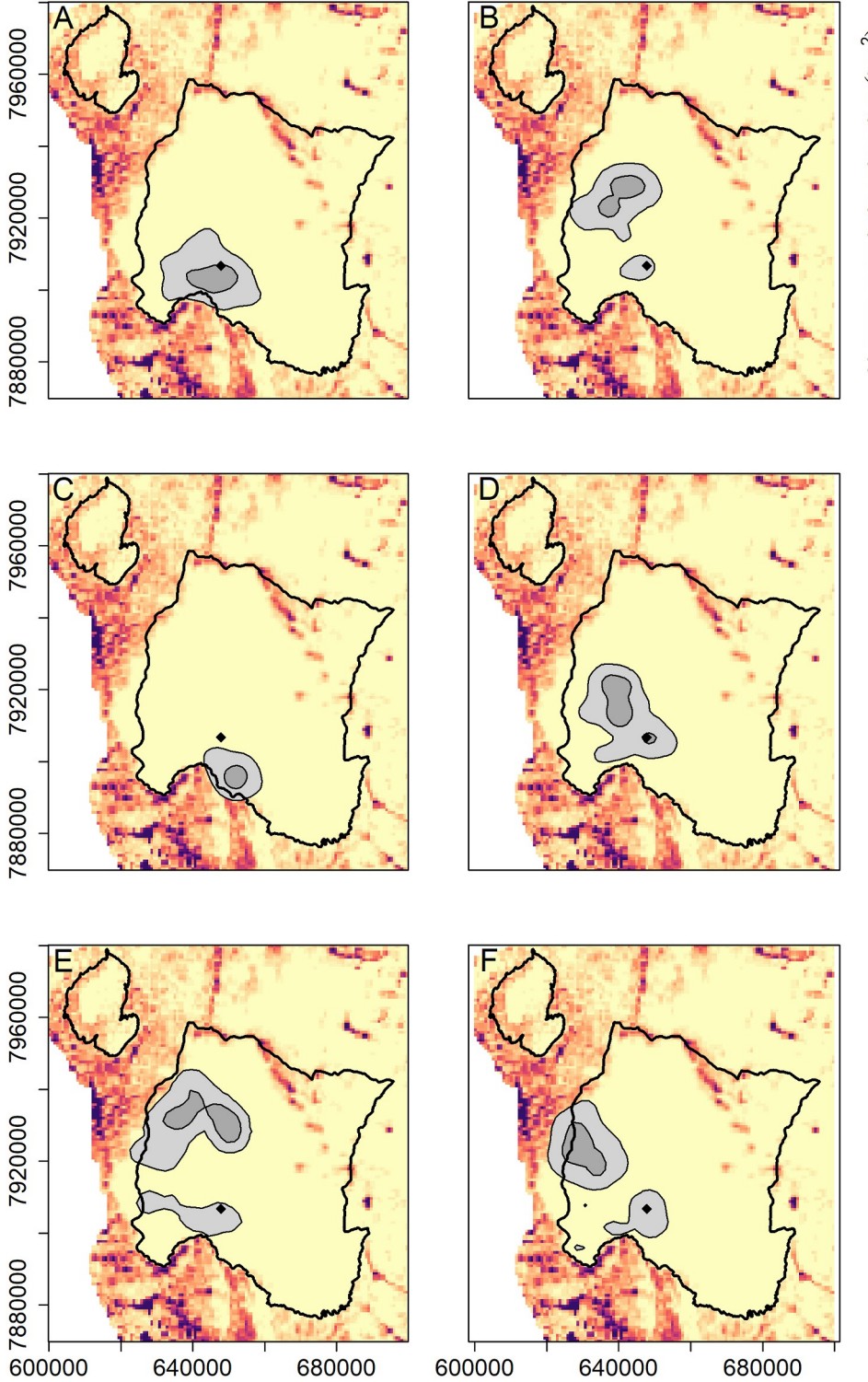

**Fig 5. The 95% total territory area (light grey) and the 50% core territory area (dark grey) for wild dogs in Gorongosa National Park (solid black line) between June 2018 and September 2020 overlaid on the human population density raster for the Gorongosa-Marromeu landscape.** Panels represent different groups of wild dogs: (A) Gorongosa pack, (B) Pwadzi pack, (C) Cheza pack, (D) Mopane pack, (E) Mucodza dispersal group and (F) Xivulo dispersal group. The diamond symbol represents the enclosure release site for the two founder packs (Gorongosa and Pwadzi packs).

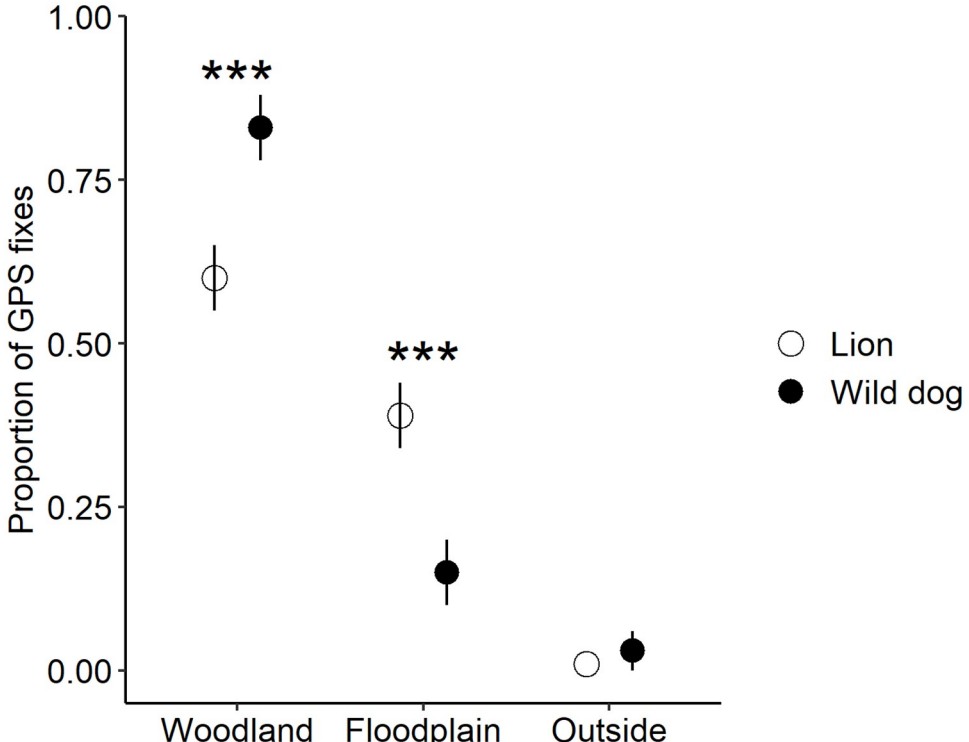

**Fig 6. The difference in woodland and floodplain habitat use by wild dogs and lions in Gorongosa National Park between June 2018 and September 2020.** Error bars represent intergroup variation in the proportion of GPS fixes while three stars indicate highly significant differences (i.e. p < 0.001) in habitat use between and within these two carnivore species.

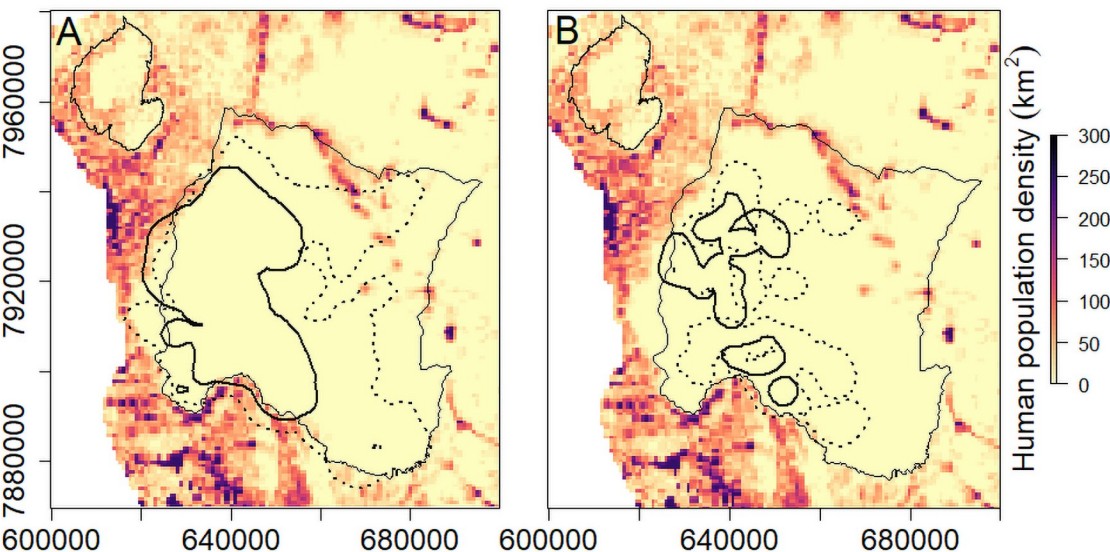

**Fig 7. Territories of wild dogs (solid line) and lions (dotted line) in Gorongosa National Park between June 2018 and September 2020 for the (A) 95% total territory areas and (B) the 50% core territory areas.** The boundary of GNP is shown as thin black line with the maps overlaid on that of human population density for the Gorongosa-Marromeu landscape.

## Discussion

The translocation of wild dogs to GNP represents the first attempted reintroduction of wild dogs into Mozambique and only the second reintroduction into an unfenced system on the continent [18]. Prior published wild dog reintroduction efforts have focused on small, fenced reserves in South Africa [15, 16]. Our study results provide important insights into the potential for successful reintroductions of wild dogs into other unfenced ecosystems on the continent. A combination of technological approaches such as GPS collars and camera traps as well as intensive *in-situ* field monitoring provided the high resolution, baseline data necessary to understand recovery dynamics and evaluate reintroduction success over the short term; this baseline is also crucial for long-term monitoring. Insights gained contribute not only to the advancement of the science of reintroduction biology but also help provide a practical, evidence-based framework to be employed by protected area managers [18, 37].

The reintroduction of wild dogs to GNP has unequivocally been successful over the short-term (i.e. the first 28 months); all five indicator categories and all nine indicators (Table 1) were achieved. Both reintroduced packs bonded and adjusted well to their new environment, and original founder members remained within their packs throughout the study period (indicator 1A, Table 1). Both packs established alpha pairs (indicator 1B, Table 1). Considering that the pack is the functional unit in wild dog society, the formation of an alpha pair [48] coupled with strong group cohesion [45, 46] in pre-release enclosures, are critical for successful pack formation. Thus, future reintroduction efforts (into fenced or unfenced areas) should place high priority on bonding of artificial packs and acclimating already formed packs in pre-release enclosures (i.e. a soft release approach) as these pre-release procedures have important impacts on fitness [47]. Intensive daily monitoring at the pre-release enclosure allowed us to assess individual dynamics to confirm pack cohesiveness, alpha pair establishment and habituation to their new environment. While we highly recommend the soft-release approach, this process needs to be coupled with intensive monitoring that enables practitioners to evaluate the success of such bonding and acclimatization attempts prior to release.

The long-term goal of a reintroduction is the establishment of a self-sustaining population with minimal further management needed [40]. For wild dogs, this would be evident by natural dispersal resulting in new pack formation events. Over our study period, wild dogs in GNP dispersed (indicator 2A) and formed new packs naturally (i.e. without human intervention, indicator 2B). This, coupled with breeding by the founder-generation (indicator 3A), high survival rates (indicators 4A and 4B) and recruitment of pups (indicator 4B), resulted in a growing population of wild dogs (indicators 5A and 5B). The key factors that should be met in order to classify the reintroduction of wild dogs as a success over the short-term is breeding and survival of the release generation [37]. Thus, overall the reintroduction has been successful over the short-term and future reintroduction attempts in other systems need to consider breeding and survival as fundamental metrics of short-term success. Notable is that individual survival rates were equal to the highest ever documented for wild dogs across their distribution [44, 65].

Breeding in wild dog packs is normally monopolized by a dominant alpha pair [44, 100]. However, we observed multiple females per pack breeding more often than only alpha breeding. Shared maternity in wild dogs is rare [65, 100, 101], and while we acknowledge our small sample size (five females in two packs, across two years), breeding subordinate females produced 19 more pups than would have been expected from alpha only breeding. This likely had important effects on population size and is an important, yet unusual, characteristic of this reintroduced population. We tentatively suggest that the high prey availability in GNP [20]

and low inter- and intra-specific competition [25] facilitated this novel breeding pattern and that bolstered pup recruitment.

Despite all packs having bred over the study, the Gorongosa pack did not succeed in raising pups in their first year. This failed breeding event in 2018 could be attributed to predation by a python residing at the den (S2 Table) but could also potentially be explained by the timing of the release of the pack from the enclosure. The pack was released in mid-June, which is typically halfway through a normal breeding season for wild dog populations [44, 68, 84]. Considering that the alpha female was observed to be pregnant prior to release and a short gestation period for wild dogs [102], the pack may not have had adequate local experience to select a more secure den site, which is essential in ensuring pup production and survival [103]. We suggest that future reintroductions allow for sufficient post-release acclimation for a pack preparing to den, otherwise this could compromise breeding.

A likely important stimulant for natural dynamics was the introduction of the second pack (i.e. Pwadzi pack); this release stimulated dispersal (indicator 2A) from the Gorongosa pack which in-turn stimulated dispersal within the Pwadzi pack which resulted in natural pack formation (indicator 2B) of a fourth pack in GNP. In small fenced areas like the metapopulation in South Africa, single packs and their dispersing cohorts struggle to find mates (mate-finding allee effect) which can constrain population growth [37, 49, 66]. In GNP, introducing a second pack resulted in three quick dispersal events (Fig 2) which closely simulated natural population dynamics. This mimicry of natural dynamics during this early stage is promising given the long-term objective of recovery programs utilizing reintroductions is to phase out further interventions with ideally all dynamics being natural and the population being self-sustaining [37, 40]. Thus, the introduction of viable alternate mates during wild dog reintroductions is a critically important aspect for recovering a population of this endangered carnivore.

Space use and prey selection are important behaviors that can explain the success of this reintroduced population. Wild dogs in GNP effectively avoided areas (woodland habitat and core lion territories) that are intensively used by lions (i.e. spatial partitioning, [44, 54–56]) which allowed them to live sympatrically in the landscape with their dominant natural threat [33]. Additionally, wild dogs also preferred different prey species than lions did in GNP (i.e. dietary niche partitioning [57, 58]) Wild dogs and lions overlapped in their selection of waterbuck but wild dogs exhibited less of a preference based on sex or age, i.e. dogs preyed more broadly on both sexes and across ages compared to lions that had a more narrow selection of younger waterbuck [57]. The aspects of dietary niche partitioning could have reduced the likelihood of interference competition between wild dogs and lions [104] and further reduced the potential for fatal encounters for wild dogs. Another potential explanation for the apparent small influence of lions on wild dogs could be the population density of lions and vegetation density. In areas where lion density has increased [53] and habitat structure is open [57], competitive exclusion can occur and wild dogs can become extirpated from a system. The inclination for reintroduction practitioners could be a reluctance to reintroduce wild dogs to areas that have established populations of lions. However, there are exceptions where reintroduced wild dogs have thrived despite high densities of lions [65]. Thus, despite the direct threat of lions to wild dogs, future reintroduction attempts for wild dogs should consider the co-evolutionary history of both predators that is an important aspect of establishing a natural population in the long-term [40].

Wild dogs in GNP also avoided anthropogenic threats. Despite brief forays (<3 days) into high density human population areas, wild dogs tended to avoid human settlements and we recorded no human-related conflicts or mortalities. As direct and indirect human-related injury or mortality of wild dogs both inside and outside protected areas [33, 105] can severely undermine the effectiveness of a reintroduction, it is likely that avoidance of humans by wild

dogs also assisted in the success of the reintroduction. In GNP, park management trains and oversees a 264-person team of National Administration of Conservation Areas wildlife rangers that are strategically deployed across the Gorongosa-Marromeu landscape to minimize human threats to wildlife. A specialized carnivore-focused unit of rangers secure areas utilized by carnivores across this landscape (Fig 1), conducting proactive snare sweeps to reduce the likelihood of wild dog by-catch in poachers' snares. This resulted in reduced (in this case zero) incidental injury or death to wild dogs via proactive management. Thus, the reintroduction of wild dogs to unfenced landscapes that are also inhabited by people require an active effort by specialized rangers to reduce the likelihood of incidental snaring that could result in additive mortality in the population. Additionally, GRP's human-wildlife coexistence team worked closely with traditional leaders and communities around GNP to share information and increase understanding towards fostering stewardship and ownership of wildlife by citizens; they also work to prevent or mitigate conflict (alert communities of the presence of carnivores, build predator-proof enclosures, deploy rangers to ensure safety for people and stray wildlife). Prior to the release of both packs, teenagers from eco-clubs in the buffer-zone visited the park and led naming ceremonies for the wild dogs, thereby also enhancing communities' personal investments in stewardship of the wild dogs. Undoubtedly, underpinning the early successes of this reintroduction program is GNP's overall management strategy that ensures security for both the ecosystem and wildlife, and pairs that strongly with community engagement in the stewardship of such resources.

Our study demonstrates that GNP can sustain further wild dog population expansion and growth given that wild dogs have only utilized 32% of the park to-date. Considering that four packs and two dispersal groups currently occupy 32% of the park, by extrapolating this based on the possibility of wild dogs using 100% of the park, we estimate that GNP could sustain a total of 12 packs and six dispersal groups (assuming the same level of resource and competition distribution, and territory size in the remaining 68% of the park). GNP management will need to decide on the methodology with which to fill vacant habitat with wild dogs, including (1) waiting for natural colonization by the currently existing (reintroduced) population, or (2) reintroducing more wild dogs, or (3) a combined strategy. Next steps will include developing population projections for GNP that account for demographic and environmental stochasticity and determining what the optimal population size could be warranting no further reintroductions, thus achieving the long-term goal of reintroductions [40]. We suggest that if additional reintroductions in GNP are to occur then focus should be on the eastern and northern parts of GNP because: (1) these are further from the highest human population density areas, (2) they are currently vacant areas with regards to wild dogs and (3) occupation of these areas could stimulate range-expansion eastwards into the greater Gorongosa-Marromeu landscape via adjacent corridors.

The opportunity for range expansion of wild dogs across the greater Gorongosa-Marromeu landscape (Fig 1) is clear. While an indigenous population of wild dogs historically occurred in the eastern reaches of Gorongosa-Marromeu (area "D" in Fig 1; J. Andre, unpublished report), recent camera-trap surveys and field surveys by GNP have confirmed only a single remnant pack of five individuals (one female and four males) remain, and they are now highly vulnerable to extinction (P. Bouley, unpublished data). In 2018, GNP assumed direct management of a 2,800 km$^2$ sector of this larger landscape adjoining the park and has since bolstered security for wildlife (including for wild dogs) and habitat in the region. Expanding wild dog reintroduction and population recovery efforts across this greater landscape towards the goals of bolstering genetic diversity and the number and size of free-ranging packs will be critical to self-sustaining survival of the species [40] offering a stark contrast to the declining trend of populations across Africa [3]. While the priority for restoration of wild dog populations should focus

on natural colonization through the security of protected areas and establishing corridors linking such areas, this study demonstrates that reintroductions remain a viable tool for the species' conservation [40], even in large unfenced systems. With multiple recoverable landscapes available to wild dogs across southern Africa [26], this project provides a compelling framework for future reintroductions of the species with tangible conservation benefits. In GNP, the data and results we present represent the baseline for evaluating the long-term success of this reintroduction in its attempt to reestablish a self-sustaining population of wild dogs.

## Supporting information

**S1 Table. Wild dog and lion collars.** The number of fixes per wild dog and lion group used in the spatial analyses. The collar coverage dates represent the full time period from group release from enclosure/pack formation except for Cheza pack that formed in March 2019 (Fig 2) but was only GPS collared in August 2019. For the lion data, we only included data from the date of the release of the first pack of wild dogs on 15 June 2018.
(DOCX)

**S2 Table. Birth details.** Pregnancies and confirmed litters for eight female wild dogs in Gorongosa National Park between June 2018 and September 2020.
(DOCX)

**S3 Table. Group association dynamics.** Details of events that resulted in individual founder wild dogs changing their group association in Gorongosa National Park from the time of the first reintroduction in June 2018 until September 2020. AM = adult male, YF = yearling female and AF = adult female.
(DOCX)

**S4 Table. Wild dog and lion kills.** Observed number and proportion of total kills made by wild dogs and lions in Gorongosa National Park between June 2018 and September 2020. Jacob's index of selection is also shown where positive values represent preferred while negative values represent avoided.
(DOCX)

**S1 Fig. Kills and wild dog GPS fixes.** The location of (A) kills for wild dogs (black dots, n = 72) and kills for lions (red dots, n = 89) and (B) wild dog GPS fixes (n = 2,985) in Gorongosa National Park between June 2018 and September 2020, overlaid onto vegetation type (woodland–grey; floodplain–green), also showing Lake Urema (blue).
(TIF)

**S2 Fig. Pack den sites.** Den sites (black diamonds) used by packs of wild dogs in Gorongosa National Park encompassing the 2018, 2019 and 2020 denning seasons overlaid onto the broad vegetation categories (woodland–grey; floodplain–green) and showing Lake Urema (blue). Panels represent different packs, specifically: (A) Gorongosa pack, (B) Pwadzi pack, (C) Cheza pack and (D) Mopane pack with pack specific territory areas shown as solid black lines, representing the 95% total territory area (outer black lines) and the 50% core territory area (inner black lines).
(TIF)

**S3 Fig. Wild dog range expansion.** Total areas used by reintroduced wild dogs in Gorongosa National Park during the first 14 months of the study (June 2018 –July 2019, dashed black line) compared to the second 14 months of the study (Aug 2019 –Sep 2020, solid black line). The location of the release enclosure (diamond), Lake Urema (blue), and broad vegetation

categories (woodland–grey; floodplain–green) are also shown. The total areas represent the merged 95% kernel UD isopleths from each pack per half of the study period.
(TIF)

**S4 Fig. Area expansion for wild dogs.** The monthly area size used by the reintroduced wild dogs between June 2018 and September 2020. Area size was delineated by the 95% minimum convex polygon enclosing all GPS fixes for all wild dogs combined.
(TIF)

**S1 Appendix. Den observations.** Interesting observations at wild dog den sites in 2019 and 2020 in Gorongosa National Park.
(DOCX)

## Acknowledgments

We want to thank management of Gorongosa National Park and Mozambique's Association of National Conservation Areas (ANAC) for approving this reintroduction and the subsequent research. We also thank Ezemvelo KZN Wildlife, Khamab Kalahari Game Reserve and the North West Provincial Government in South Africa for donating the founder wild dogs to the project. We also thank WildlifeACT in South Africa for logistical, technical and veterinary support in this project and the invaluable support of the Bateleurs for providing the charter planes and pilots to fly the wild dogs in both reintroduction phases. We thank the team of GNP wildlife rangers for continuous safe-guarding of the new population, and media teams, safari guides and visitors in the park for submitting valuable wild dog sightings to the project. A very special thanks to the Carr Foundation, Oak Foundation and National Geographic for their support of wild dog recovery and conservation in Mozambique.

## Author Contributions

**Conceptualization:** Paola Bouley, Cole Du Plessis, David G. Marneweck.

**Data curation:** Paola Bouley, Mercia Angela.

**Formal analysis:** Paola Bouley, David G. Marneweck.

**Funding acquisition:** Paola Bouley.

**Investigation:** Paola Bouley, Antonio Paulo, Mercia Angela, David G. Marneweck.

**Methodology:** Paola Bouley, Antonio Paulo, Mercia Angela, Cole Du Plessis, David G. Marneweck.

**Project administration:** Paola Bouley.

**Resources:** Cole Du Plessis.

**Software:** David G. Marneweck.

**Supervision:** Paola Bouley.

**Validation:** Paola Bouley, David G. Marneweck.

**Visualization:** David G. Marneweck.

**Writing – original draft:** Paola Bouley, David G. Marneweck.

**Writing – review & editing:** Paola Bouley, Antonio Paulo, Mercia Angela, Cole Du Plessis, David G. Marneweck.

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
