## [Decision Letter · Decision Letter 0]

29 Dec 2020

PONE-D-20-35800

Call of the wild: the successful reintroduction of African wild dogs to Gorongosa National Park, Mozambique

PLOS ONE

Dear Dr. Bouley,

Thank you for submitting your manuscript to PLOS ONE. After careful consideration, we feel that it has merit but does not fully meet PLOS ONE’s publication criteria as it currently stands. Therefore, we invite you to submit a revised version of the manuscript that addresses the points raised during the review process.

Your manuscript has the potential to make an excellent contribution to the literature of carnivore ecology and conservation generally and more specifically on their reintroductions. Your submission has received positive feedback as well as helpful suggestions for improvement from three reviewers with expertise in carnivore ecology, wild dog ecology and conservation, and the recovery of Mozambique's wildlife. The major area of divergence in the reviews surrounds the utility and presentation of the recovery indicators. Although I agree with Reviewer #3 that the manuscript is lengthy and could be more focused, I think an indicator or success criteria table could be useful for future reintroduction efforts and therefore may be worth keeping, but perhaps following revisions based on suggestions from the reviewers, including the minor detail of more clearly enumerating the criteria used. Below (and attached) you will find helpful comments from three reviewers to assist you with revisions.

We look forward to receiving your revised manuscript.

Kind regards,

Stephanie S. Romanach, Ph.D.

Academic Editor

PLOS ONE

Journal Requirements:

3. We note that Figures 1, 6 and S1-S3 in your submission contain map/satellite images which may be copyrighted.

a. You may seek permission from the original copyright holder of Figures 1, 6 and S1-S3 to publish the content specifically under the CC BY 4.0 license. 

5. Please amend the manuscript submission data (via Edit Submission) to include author Cole Du Plessis.

Additional Editor Comments:

- Given that you intend for others to be able to utilize the information you present for future translocations, it would be helpful for you to include a brief statement in your Ethics section on how translocations were conducted (e.g., in the presence of a vet, use of blindfolds, ear muffs), and perhaps include additional details (e.g., collar weight, drug combinations) as Supplementary material.

- L 591 and 595: "predate" should be corrected to "prey"

Reviewers' comments:

Reviewer's Responses to Questions

**Comments to the Author**

1. Is the manuscript technically sound, and do the data support the conclusions?

Reviewer #1: Yes

Reviewer #2: Yes

Reviewer #3: Partly

2. Has the statistical analysis been performed appropriately and rigorously? 

Reviewer #1: Yes

Reviewer #2: I Don't Know

Reviewer #3: N/A

3. Have the authors made all data underlying the findings in their manuscript fully available?

Reviewer #1: No

Reviewer #2: Yes

Reviewer #3: Yes

4. Is the manuscript presented in an intelligible fashion and written in standard English?

Reviewer #1: Yes

Reviewer #2: Yes

Reviewer #3: Yes

5. Review Comments to the Author

Reviewer #1: This study by Bouley et al. is a thorough accounting of the successful reintroduction of endangered African wild dogs into Gongorosa National Park (GNP), Mozambique. GNP is rapidly becoming a paradigm for the large-scale, whole-ecosystem restoration, and carnivores remain the critical "missing link" in restoring a natural regime of ecosystem processes. As such, I consider this study is a very important contribution to the emerging literature on rewilding; this work serves both as an essential puzzle piece in the larger story of a large-scale rewilding project with few if any parallels worldwide AND as a step-by-step procedural guide to translocation and monitoring that will be practically useful for other estates that seek to (re)introduce wild dogs (or wolves, or other large carnivores for that matter). The authors outline a 16 (or 17, see below) point framework by which they benchmarked the success of the initiative, which is helpfully systematic and generalizable to other circumstances. On the whole, the work seems to have been carefully and rigorously done and is thoroughly described in the manuscript; I have some specific suggestions for improvement below, but pending the authors responses to these, I think that this study meet's PLOS ONE's criteria for publication and I would strongly advocate its publication once these and other reviewers' comments have been addressed.

28: Instead of "carnivores", I would go with "these species", because the statement ("are threatened globally") really applies to large ("top") mammalian carnivores, as opposed e.g. other members of Carnivora (e.g., raccoons, skunks, domestic cats and dogs, etc.) or predatory animals more broadly.

42-43: "First successful reintroduction of wild dogs into an open landscape" -- there have been many translocated dogs and an extensive literature on it, so this statement seems to hinge on the definition of "open" -- do you mean unfenced?

73: I would add citation to Seddon et al. 2014. Reversing defaunation: Restoring species in a changing world. Science 345:406-412

79-80: Direct persecution of wild dogs by ranchers and pastoralists has been common throughout much of the continent, and seems like a curious omission here where listing drivers of population decline.

99: In the context of this study, I think it's important to note here that recent work shows that the incompleteness of the carnivore guild has left an enduring ecological imprint on Gongorosa NP, per the recent study by Atkins et al. 2019. Cascading impacts of large-carnivore extirpation in an African ecosystem Science 364:173-177).

Table 1: It is a little confusing that the Abstract refers to 16 indicators, but the Table shows 10 categories comprising what appear to be 17 specific indicators (if I count correctly). Also, although a bit premature in that the results have not yet been presented, I think it would be helpful to include a column indicating if the indicator was achieved (which is yes for almost all of them, but would be helpful to have this as a quick look-up guide for those who might be skimming the paper).

137: "current" -- I would specify what year you mean by this, as this is otherwise ambiguous. A reference for this sentence would be helpful.

165: What other resident populations in the region are known?

170-171: This is important information, but what is the reference point for these reductions in poaching? I.e., in 2018 as compared with what?

193-194: There is some grammatical hiccup in this sentence "reconnecting GNP to any other confirmed and no resident populations". Cf. my comment on line 165.

273: A few more details on husbandry would be helpful, at least from my perspective. Were whole or butchered prey items offered? Do you know approximately what fraction of the prey were impala vs. waterbuck? Did dogs feed equally well on all foods offered? These details are not strictly essential but may help guide others attempting to follow your example.

354: Can you be certain you detected *all* successful hunts? Seems unlikely to me.

356: How was "prey biomass" defined? E.g., did this include all of the species included in the aerial counts, including e.g. elephant and hippo that wild dog rarely if ever take down as well as species that might be too infrequent in GNP to be common prey? Or did you define a subset of counted species that are likely (in terms of size and relative abundance) to be eaten by wild dogs? Or did you only include the species that were actually observed to have been killed by wild dogs during this study as potential prey? These details are important for interpreting the selectivity measurements described in the next sentence. From Fig. 4, it seems that you only deemed a subset of counted species as potential prey, but specifying your criteria here is important. From Table S4, the dogs never ate a hartebeest, wildebeest, sable, or buffalo (Table S4), but only the latter three species are shown in Fig. 4. Which species were defined as potential prey or not will influence estimates of selectivity by influencing the inferred relative abundances of potential prey spp.

357-363: Similar to previous comment, a bit more detail about this selectivity analysis would be helpful -- most notably how you defined availability, but also how you converted your kill observations into data on utilization by dogs. It's not obvious to me that these calculations could be independently replicated without a bit more supporting information.

460-465: These were non-parametric regressions? I think the test statistics output from lm() in R are all based on parametric assumptions. Can you clarify?

491: Did the fight result in injury?

589: As noted above, I find it hard to evaluate the selectivity results based on the methods provided. More generally, although the selectivity data are important and interesting, they do not tell a complete story owing to the heavily skewed relative abundances of the prey species. I would include a 2nd panel in Fig. 4 that just shows the breakdown of prey by species in the form of a bar graph or pie chart; these data are presented in full in Table S4, but a graphical summary would be a helpful addition to the main text. I would also qualify the word "avoiding" on line 589 with "in relation to their availability", just to minimize the potential for confusion. Wild dogs ate a lot of waterbuck, which is interesting and perhaps surprising; that they were avoided (in relation to availability) seems to reflect the disproportionate relative abundance of waterbuck relative to all other prey (as noted elsewhere in the text).

Related: Was there any difference between packs in prey utilization (or selectivity)? Some experts believe that wild dogs have a strong sense of "tradition" in their prey preferences, and wondering if you saw any evidence of that as reflected in differences between packs.

587-600 and throughout: I would use past tense consistently in describing results.

609: This is a 117% increase (100% increase being a doubling), not a 217% increase.

647: Descriptive stats on absolute number of kills in each habitat type would be helpful as a complement to (and arguably more important than) the statistical test. In Fig. S2, the red dots for lions eclipse the black dots for wild dogs, making it hard to discern where the wild dog kills occurred. As the focus of this paper is on wild dogs, I think it would be more helpful to bring the black dots to the foreground. Similarly, making this figure larger (e.g., by stacking the panels vertically instead of horizontally and expanding to fill the page margins) would be helpful.

701-702: The prey data suggest that wild dog and lions are also avoiding each other through differential foraging patterns/prey selection (cf. parts of introduction about interference competition).

707-708: A few more words about what this competition at dens looked like? Any observations of interactions between packs (or between WD and lion)? Ah -- never mind, I see this information in S1 Appendix. You should call that out here.

Discussion: Some brief comment on patterns of prey use/selection in relation to the larger literature on wild dog foraging would be helpful. It is surprising (to me, at least) that although wild dog feast on impala in much of their range, impala are abundant in GNP, and impala were among the feeds used in the boma, the dogs ate very few impala after being released.

Data Availability: I do not see a Data Availability Statement. Authors should include a statement affirming their adherence to PLOS data availability requirements (https://journals.plos.org/plosone/s/data-availability) and deposit these data within a repository or as supporting information files. This is important for maximizing reproducibility and ultimate scientific and conservation impact of the research.

Reviewer #2: This paper has largely been a pleasure to read. It’s well written and informative, and gives sufficient detail without being overwhelmingly bogged down in unnecessary statistics, which makes it very accessible to readers. It also represents a short-term success story for reintroduction of African wild dogs into an open ecosystem which is a valuable contribution to the literature (and to conservation). I have recommended it for publication with a few minor changes.

Most of my comments are on the PDF. A few key points are expanded below.

I would like to see a little more explanation of the due diligence conducted to ensure the genetic suitability of the reintroduced stock (i.e. your process of determining that it was suitable to bring dogs from SA into Mozambique).

The Breeding section in the Results needs re-writing slightly. Please see my comments. It’s not incorrect, just very confusing (I spent a while wondering how there could have been 11 pregnancies in each year!!).

I think the discussion could be a bit fuller. You observe some very interesting things but given them no space in the discussion. This is very focused on the success of the reintroduction, which is valid, being the purpose of the paper, but please see my comments in the attached PDF asking for a little more interpretation of some of the more interesting / important findings.

Reviewer #3: Comment to manuscript PONE-D-20-35800 “Call of the wild: the successful reintroduction of African wild dogs to Gorongosa National Park” by Bouley and colleagues.

The authors report of a successful reintroduction attempt of African wild dogs to Gorongosa National Park, from where wild dogs have been absent since the civil war. The authors identify 16 criteria that they deemed appropriate to assess reintroduction success. The manuscript is well written and this is a success story that needs to be told. There is an incredible amount of very meticulous work that was put during fieldwork and in the preparation of the manuscript.

The manuscripts mainly describes observations and data collected in a rather detailed and reliable manner and I do not have major comments. This meticulousness in describing the events is, however, a double-edged sward:

1) The manuscript is incredibly long, instead of identifying a couple key processes that are really relevant to assess reintroduction success. In its present form the manuscript is not focused. Please refer to comment below re some of the 16 criteria.

2) This is one of the few successful reintroduction events (despite not the first; see for example Jackson et al. 2012 Wildlife Research) in an unfenced area. This makes it even more important to discuss why this attempt was successful and why others were not. Differences in release method? Differences in presence/abundance of other carnivores? Differences in habitat characteristics? Differences in prey availability? Unfortunately there is no mentioning of all these in the discussion. On this note, it is important to note that a large unfenced area is probably not very different from an unfenced one in the first phases of reintroduction, of course the situation is different for some fenced parks in SA that are smaller than the size of an average wild dog pack home range.

3) In my opinion, few elements are key and stand out and I suggest focusing on these instead of trying to report on everything that has been collected. These are the topic in the result section under “breeding”, survival”, “population size”, and “territory size and overlap”. For instance the high number of subordinate females reproducing is unusual for any well-established natural population.

4) I do not agree with many of the 16 criteria (see below) but I also think that the authors should move away from those. That the reintroduction was successful is irrefutable, the authors do not need 16 (questionable) criteria to prove that. Instead they need to say why this attempt was successful and what we should learn from this. Under which circumstances can we expect reintroduction to work and under which reintroduction may not work? This is the lesson that conservation needs to take home

A few general comments to the introduction and to the 16 categories

- the first paragraph appears inappropriate and too generic. I suggest deleting it completely and starting directly wit the second paragraph, which is more focused and pertinent to the topic of the manuscript.

- The paragraph starting with “The recently restored Gorongosa National Park (GNP)” should then be moved in second position and the paragraph starting with “The African wild dog (Lycaon pictus)” in third position. After this paragraph there is the need to add a paragraph describing the indicators used to assess successful reintroduction (see next comment).

- There is a total lack of hypothesis. What and why should be expected to be a good measure for reintroduction success? Which parameters should be considered? I see this info is summarized in Table 1, but I suggest a paragraph describing the broad categories used, and why these are important indicators. For instance,

1A and 1B: how does this apply to phase two since an entire pack has been translocated?

4B: why survival of 50% of founders should be used as indication of successful reintroduction? Is this because this matches survival rate of founder individuals in the wild? Positive pop growth depends on mortality (of founders) and birth, not only on mortality of founders.

5A: what shall be understood with “population size increases significantly”, what has to be considered a “significant” increase and why?

6A: I disagree. Diet has nothing to do with successful reintroduction. What if reintroduced dogs specialize on small rodents (I know, silly example) and still thrive? Would you consider reintroduction to be unsuccessful?

7A: is a trivial consequence of 2B, so redundant.

8A: Again, there is no rational to use prey/diet as a metric to assess successful reintroduction.

8B: Makes no sense. I know that lions are among the main causes of mortality in wild dogs, but what if the reintroduced dogs, despite using areas used by lions, thrive and adults and pups survive? Would you consider this as a failed reintroduction? Important is to quantify lions density to allow comparison with other areas where reintroduction has also been attempted. But I don’t see the need to calculate overlap or avoidance.

9A: similar as for 8B

9B: It is irrelevant of what they die of, as long as 4B and 5A apply. Also, how is “minimal” mortality quantified? Additionally, the authors state that “threats that the lead to the historical extirpation of wild dogs in GNP have been alleviated (l. 172). So this means that one of the criteria to assess successful reintroduction applies is even before reintroduction starts…

10A and 10B: Makes little sense. For instance, it is up to the Restoration Project managers to decide who should oversee the reintroduction. So basically by deciding if they want to hire “wildlife veterinarians of

national origin” or “wildlife veterinarians from abroad” they indirectly decide (according to your categories) whether the reintroduction is successful or not?

There is indeed the need to justify the choice of these categories! Or better so, to re-think the structure of the manuscript and move away from the criteria.

I hope my comments will help in focusing the manuscript on key aspect of reintroduction, thus rendering it shorter and more accessible.

6. PLOS authors have the option to publish the peer review history of their article (what does this mean?). If published, this will include your full peer review and any attached files.

Reviewer #1: No

Reviewer #2: No

Reviewer #3: No

---

## [Author Response · Author response to Decision Letter 0]

17 Mar 2021

Thank you for the opportunity to respond. We have uploaded the rebuttal/response letter as requested in the Decision Letter.

---

## [Editor Report · Decision Letter 1]

26 Mar 2021

The successful reintroduction of African wild dogs (Lycaon pictus) to Gorongosa National Park, Mozambique

PONE-D-20-35800R1

Dear Dr. Bouley,

We’re pleased to inform you that your manuscript has been judged scientifically suitable for publication and will be formally accepted for publication once it meets all outstanding technical requirements.

Kind regards,

Stephanie S. Romanach, Ph.D.

Academic Editor

PLOS ONE
---

## [Editor Report · Acceptance letter]

6 Apr 2021

PONE-D-20-35800R1 

The successful reintroduction of African wild dogs (*Lycaon pictus*)
to Gorongosa National Park, Mozambique 

Dear Dr. Bouley:

I'm pleased to inform you that your manuscript has been deemed suitable for publication in PLOS ONE. Congratulations! Your manuscript is now with our production department. 

Kind regards, 

on behalf of

Dr. Stephanie S. Romanach 

Academic Editor

PLOS ONE